# Response of Two Major Lakes in the Changtang National Nature Reserve, Tibetan Plateau to Climate and Anthropogenic Changes over the Past 50 Years

Zhilong Zhao [1,2] , Zengzeng Hu [3,*], Jun Zhou [1], Ruliang Kan [1] and Wangjun Li [4]

1   College of Economics and Management, China Three Gorges University, Yichang 443002, China
2   Faculty of Geographical Science, Beijing Normal University, Beijing 100875, China
3   Beijing Academy of Sciences and Technology, Beijing 100089, China
4   School of Geography Science and Geomatics Engineering, Suzhou University of Science and Technology, Suzhou 215009, China
*   Correspondence: huzengzeng@bjss.org.cn

**Abstract:** Areal changes in alpine lakes on the Tibetan Plateau (TP) are reliable indicators of climate change and anthropogenic disturbance. This study used long-term Landsat images and meteorological records to monitor the temporal evolution patterns of lakes within the Changtang National Nature Reserve between 1972 and 2021 and examine the climatic and anthropogenic impacts on lake area changes. The results revealed that the area of Lake LongmuCo and Lake Jiezechaqia significantly expanded by 12.81% and 12.88% from 1972 to 2021, respectively. After 1999, Lake LongmuCo and Lake Jiezechaqia entered into a period of rapid expansion. During 1972–2021, the annual mean temperature significantly increased at a rate of 0.05 °C/a, while the change in annual precipitation was not significant. The temperature change was a major contributor to the observed changes of Lake LongmuCo and Lake Jiezechaqia between 1972 and 2021, while human intervention also played a vital role during 2013–2021. The glaciers around these two lakes decreased by 21.81%, and the increase in water supply from warming-triggered glacier melting was a reason of expansion of Lake LongmuCo and Lake Jiezechaqia. The areas of the two artificial salt lakes affiliated with Lake LongmuCo and Lake Jiezechaqia were 0.24 km$^2$ and 2.67 km$^2$ in 2013 and rose to 0.51 km$^2$ and 9.80 km$^2$ in 2021, respectively. In particular, the continuous exploitations of salt lakes to extract lithium resources have retarded the rate of expansion of Lake LongmuCo and Lake Jiezechaqia. The dams constructed by industrial enterprises have blocked the expansion of Lake LongmuCo to the south. This paper sheds new light on the influences of recent human intervention and climatic variation on alpine lakes within the TP. Due to the importance of alpine lakes in the TP, we need more comprehensive and in-depth efforts to protect the lake ecosystems within the national nature reserves.

**Keywords:** lake area changes; climate change; anthropogenic activity; Tibetan Plateau; nature reserve; remote sensing monitoring

## 1. Introduction

Lakes not only play important roles in the global hydrological system and biophysical environment [1–3] but also provide vital ecosystem services for the sustainable development of the global economy [3–6]. Lake area change is a reliable indicator of climate change and anthropogenic disturbance, especially lakes in arid and cold regions [7–9]. In recent decades, lakes worldwide have shown large and widespread variations in the context of accelerated climatic change and increasing anthropogenic impacts [9–13]. Major threats to lakes include the human use of land, such as farming, mining, spatial expansion of human settlements [4,14], and climatic variation, factors that influence the area, level, and volume of lakes, especially in different latitudes [4,7,14–20]. Protecting and restoring natural ecosystems, including lake ecosystems, is widely viewed as a win–win strategy for addressing

the challenges of human influence and climate change [4,12,21]. The establishment of protected areas is a major component of ecosystem conservation [22]. Globally, there were 261,200 Protected Areas in 2019 covering about 15% of the land area [23–25].

The Tibetan Plateau (TP) is considered the "Third pole" [26] and the "Asian water tower" [27]. There are currently ~1400 lakes on the TP, covering an area of approximately $5 \times 10^4$ km$^2$ [10,28]. Lake ecosystems in the TP have been severely altered by climatic warming and human activity [8,29]. Since the 1980s, a large number of nature reserves have been established to conserve natural ecosystems, including lake ecosystems, by restricting and prohibiting human activities [25,30]. In 2020, the Three-River-Source National Park, one of the first national parks in China, was established on the TP [25]. With the implementation of constructive projects to protect the ecological security of nature reserves on the TP, human activities showed positive impacts on the alpine ecosystems. However, there are still negative impacts of anthropogenic changes on ecosystems such as lake ecosystems [25]. Investigation of the lake spatio-temporal variability on the TP is essential for understanding the current state and future trajectory of lakes, evaluating water resources, and analyzing hydrologic processes in the context of climatic variation and anthropogenic intervention [8,28]. However, the paucity of historical lake area records and the physical difficulties of accessing results have severely limited our knowledge of the spatial patterns and processes of lakes in remote areas of the TP [8,9]. Fortunately, remote sensing (RS) has been widely used to examine lake dynamics and to incorporate more factors in analyses of climate change and human impact [25,31–34].

Numerous RS studies have reported that lake expansion since the late 1990s is one of the most outstanding environmental change events on the TP [10,34–36]. The lake expansion has been accelerated since the 2000s and has primarily occurred in areas of the Inner TP, such as the Changtang Plateau [34–37], while most of the lakes in the southern and eastern TP have experienced shrinkage over the last several decades [38–45]. Generally, climate factors were the major cause for the lake area changes on the TP, especially for lakes in the endorheic TP [40–45]. Moreover, climate change effects such as increased precipitation, rising temperature, glacial melting, and permafrost degradation have contributed to the lake changes in the TP in recent years [28,34–39]. However, research on the role of human activities in the lake changes is still lacking. In particular, researchers have generally focused on the influences of climate changes to lake changes in the Inner TP. Very limited studies have been conducted on the pivotal role of human intervention in the lake changes within the endorheic TP, such as the influences of the exploitation of salt lakes in the Changtang National Nature Reserve (CNNR) on lake changes.

Here, we selected two lakes (Lake LongmuCo and Lake Jiezechaqia) in the CNNR in the hinterland of the TP. These lakes are in one of the 25 National Key Ecological Function Areas in China and are also in one of the key implementation areas of the ecological conservation and restoration project in the TP ecological barrier area [46]. Furthermore, since Lake Jiezechaqia and Lake LongmuCo have the first and second lithium resource reserves in Tibet, respectively, two artificial salt lakes have been built around them by a state-owned enterprise. After the exploitation of salt lakes, these two lakes have become the only two lakes where lithium is being extracted at present in the CNNR, and the two largest lakes in Tibet for extracting lithium resources. Employing Landsat archive data and meteorological data, this study aimed to explore the patterns of area changes of lakes from 1972 to 2021 and to analyze the climatic and anthropogenic impacts on lake area changes. As a case study, this paper provides a better understanding of lake area change and its influencing factors within the TP.

## 2. Materials and Methods

### 2.1. Study Area

Lake LongmuCo (80°27.41′ E, 34°36.85′ N, 5005 m a.s.l.) and Lake Jiezechaqia (80°54.14′ E, 33°57.26′ N, 4527 m a.s.l.) are glacier-fed lakes [10]. They are located on the western edge of the CNNR [47,48] (Figure 1). Calculated from Landsat images, the

average areas of Lake LongmuCo and Lake Jiezechaqia were 106.80 km$^2$ and 114.98 km$^2$ in the period of 1972–2021, respectively. The nearest road to Lake LongmuCo is National Highway No. 219 (Figure 1).

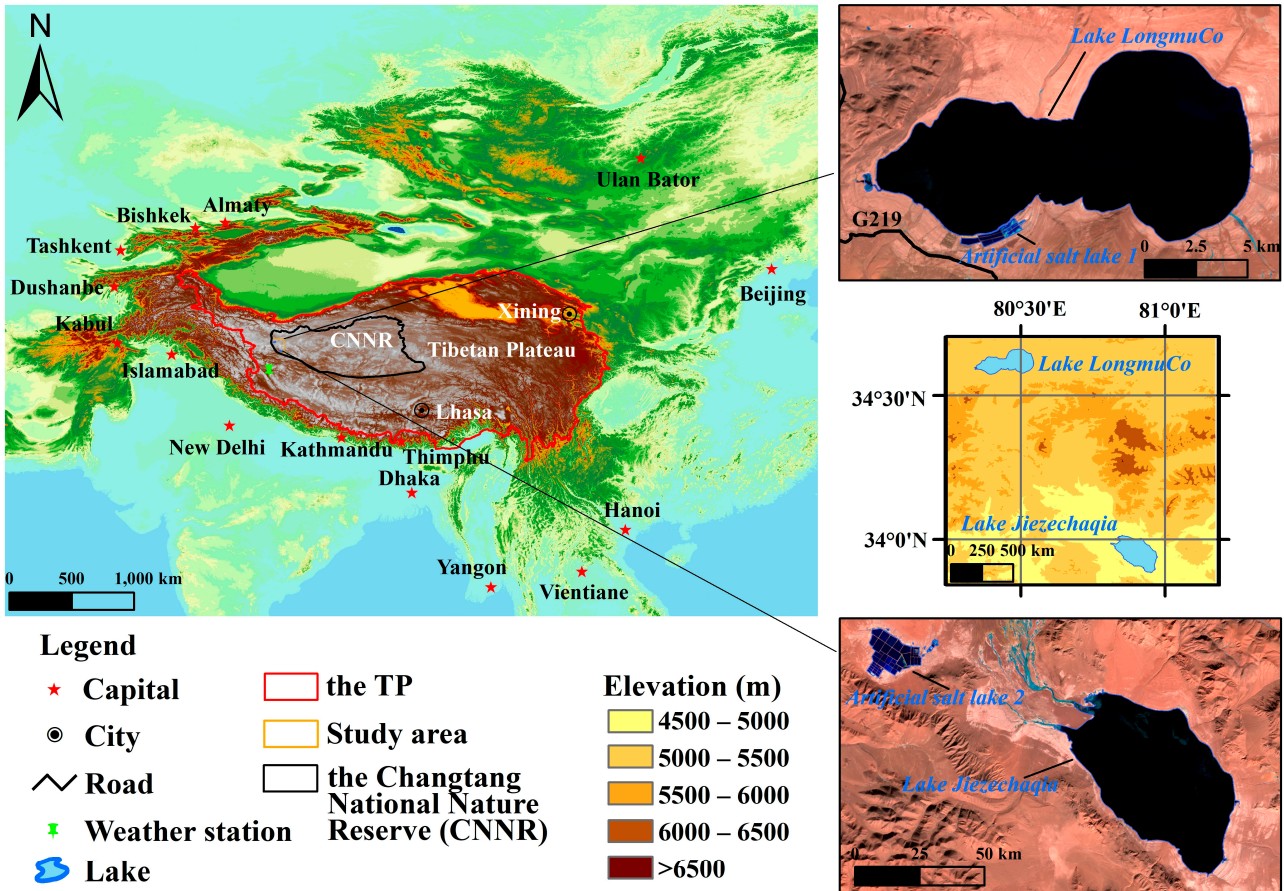

**Figure 1.** Locations of Lake LongmuCo and Lake Jiezechaqia and their surrounding topography.

CNNR, with a total area of 29.8 × 10$^4$ km$^2$, officially became a nature reserve in 1993 for protecting and restoring natural habitats, including grassland, desert steppe, wetland, and lake ecosystems. It will become a national park in the future. The climate type of the CNNR is a plateau continental climate. Annual mean temperature and annual precipitation around the areas of Lake LongmuCo and Lake Jiezechaqia are 1.06 °C and 72.88 mm, respectively [49].

### 2.2. Data and Processing

#### 2.2.1. Remote Sensing (RS) Data

For lake area mapping, a total of 30 Landsat satellite images were used in this study (Table 1). These satellite data include one Landsat 1 MSS scene, ten Landsat 5 TM scenes, ten Landsat 7 ETM+ scenes, and nine Landsat 8 OLI scenes from 1972 to 2021, downloaded from the official website of the United States Geological Survey (http://glovis.usgs.gov, accessed on 1 August 2022). Among the 30 Landsat satellite image scenes, the maximum and minimum values of cloud cover were 25.80% and 0.72% (Table 1), respectively.

Since the change rate of lake extent on the TP was less than 2% from September to December, we chose RS data (Table 1) in this period each year for comparison of the interannual variability of lake area [9,50,51]. If there were several archived Landsat images available during the period from September to December each year, we selected the one with the lowest cloud cover for lake area mapping. However, the interpretation of lake extent was not performed if there were no high-quality archived images available.

**Table 1.** RS data used to derive the lake area.

| Path | Row | Date | Name | Sensor Source | Resolution (m) | Cloud Cover (%) |
|------|-----|------|------|---------------|----------------|-----------------|
| 156 | 36 | 19 December 1972 | LM11560361972354AAA02 | Landsat MSS | 60 | 20.00 |
| 145 | 36 | 7 November 1987 | LT51450361987311SGI00 | Landsat TM | 30 | 5.11 |
| 145 | 36 | 24 October 1988 | LT51450361988298SGI00 | Landsat TM | 30 | 3.89 |
| 145 | 36 | 9 December 1993 | LT51450361993343ISP00 | Landsat TM | 30 | 4.34 |
| 145 | 36 | 25 October 1994 | LT51450361994298ISP00 | Landsat TM | 30 | 1.89 |
| 145 | 36 | 15 November 1996 | LT51450361996320ISP00 | Landsat TM | 30 | 2.56 |
| 145 | 36 | 2 November 1997 | LT51450361997306ISP00 | Landsat TM | 30 | 25.80 |
| 145 | 36 | 29 September 1999 | LE71450361999272SGS01 | Landsat ETM+ | 30 | 1.00 |
| 145 | 36 | 4 December 2000 | LE71450362000339SGS00 | Landsat ETM+ | 30 | 15.00 |
| 145 | 36 | 20 October 2001 | LE71450362001293SGS00 | Landsat ETM+ | 30 | 4.00 |
| 145 | 36 | 24 November 2002 | LE71450362002328SGS05 | Landsat ETM+ | 30 | 8.00 |
| 145 | 36 | 11 November 2003 | LE71450362003315ASN01 | Landsat ETM+ | 30 | 4.68 |
| 145 | 36 | 10 September 2004 | LE71450362004254ASN01 | Landsat ETM+ | 30 | 1.48 |
| 145 | 36 | 2 December 2005 | LE71450362005336SGS00 | Landsat ETM+ | 30 | 6.36 |
| 145 | 36 | 18 October 2006 | LE71450362006291PFS00 | Landsat ETM+ | 30 | 6.62 |
| 145 | 36 | 22 November 2007 | LE71450362007326PFS00 | Landsat ETM+ | 30 | 4.95 |
| 145 | 36 | 2 December 2008 | LT51450362008337BJC00 | Landsat TM | 30 | 18.81 |
| 145 | 36 | 3 November 2009 | LT51450362009307KHC00 | Landsat TM | 30 | 17.91 |
| 145 | 36 | 8 December 2010 | LT51450362010342KHC00 | Landsat TM | 30 | 3.67 |
| 145 | 36 | 22 September 2011 | LT51450362011265KHC00 | Landsat TM | 30 | 4.00 |
| 145 | 36 | 16 September 2012 | LE71450362012260PFS00 | Landsat ETM+ | 30 | 16.98 |
| 145 | 36 | 27 September 2013 | LC81450362013270LGN01 | Landsat OLI | 30 | 1.53 |
| 145 | 36 | 17 November 2014 | LC81450362014321LGN00 | Landsat OLI | 30 | 2.51 |
| 145 | 36 | 20 November 2015 | LC81450362015324LGN00 | Landsat OLI | 30 | 1.80 |
| 145 | 36 | 8 December 2016 | LC81450362016343LGN00 | Landsat OLI | 30 | 2.10 |
| 145 | 36 | 22 September 2017 | LC81450362017265LGN00 | Landsat OLI | 30 | 2.43 |
| 145 | 36 | 25 September 2018 | LC81450362018268LGN00 | Landsat OLI | 30 | 12.40 |
| 145 | 36 | 12 September 2019 | LC81450362019255LGN02 | Landsat OLI | 30 | 1.19 |
| 145 | 36 | 30 September 2020 | LC81450362020274LGN00 | Landsat OLI | 30 | 0.72 |
| 145 | 36 | 4 November 2021 | LC81450362021308LGN00 | Landsat OLI | 30 | 9.65 |

2.2.2. Meteorological Data

In this study, the temperature and precipitation data were obtained from the China Meteorological Data Service Center (http://data.cma.cn, accessed on 5 August 2022). Among these data, we used the annual mean temperature and annual precipitation recorded at Shiquanhe Weather Station in the period of 1972–2021, the nearest meteorological station to Lake LongmuCo and Lake Jiezechaqia (Figure 1), to analyze the environmental impacts to lake changes.

2.2.3. Glacier Data

In this study, the glacier data were obtained from the first glacier inventory dataset of China (1987–2004) and the second glacier inventory dataset of China (2006–2011) [52]. Among these data, we analyzed the glacier area changes around the regions of Lake LongmuCo and Lake Jiezechaqia by comparison between the two glacier inventory datasets mentioned above.

2.2.4. Other Auxiliary Data

In this study, a 1:100,000 scale topographic map was used as the base image for geometric correction, and it was obtained from the Resource and Environment Science and Data Center (https://www.resdc.cn, accessed on 10 July 2022).

The digital elevation model (DEM) data with 30 m × 30 m spatial resolution were also obtained from the Resource and Environment Science and Data Center (https://www.resdc.cn, accessed on 2 August 2022), and the DEM data were used in drawing Figure 1.

The boundaries of the TP and the CNNR were obtained from Zhang et al. [53] and Zhang et al. [48], respectively, and these boundaries were also used in drawing Figure 1.

Lake volume change data were obtained from Zhang et al. [54] for analyzing volume changes of lakes within the study area. The lake area data were obtained from Zhang et al. [10] for data comparison.

### 2.2.5. Data Preparation

Using ENVI 5.3 software, the data preprocessing (Figure 2) of Landsat satellite images was divided into two steps: (1) geometric correction and (2) radiometric calibration and atmospheric correction. In Step (1), the value of root mean square error (RMSE) of each image was less than 0.5 pixels.

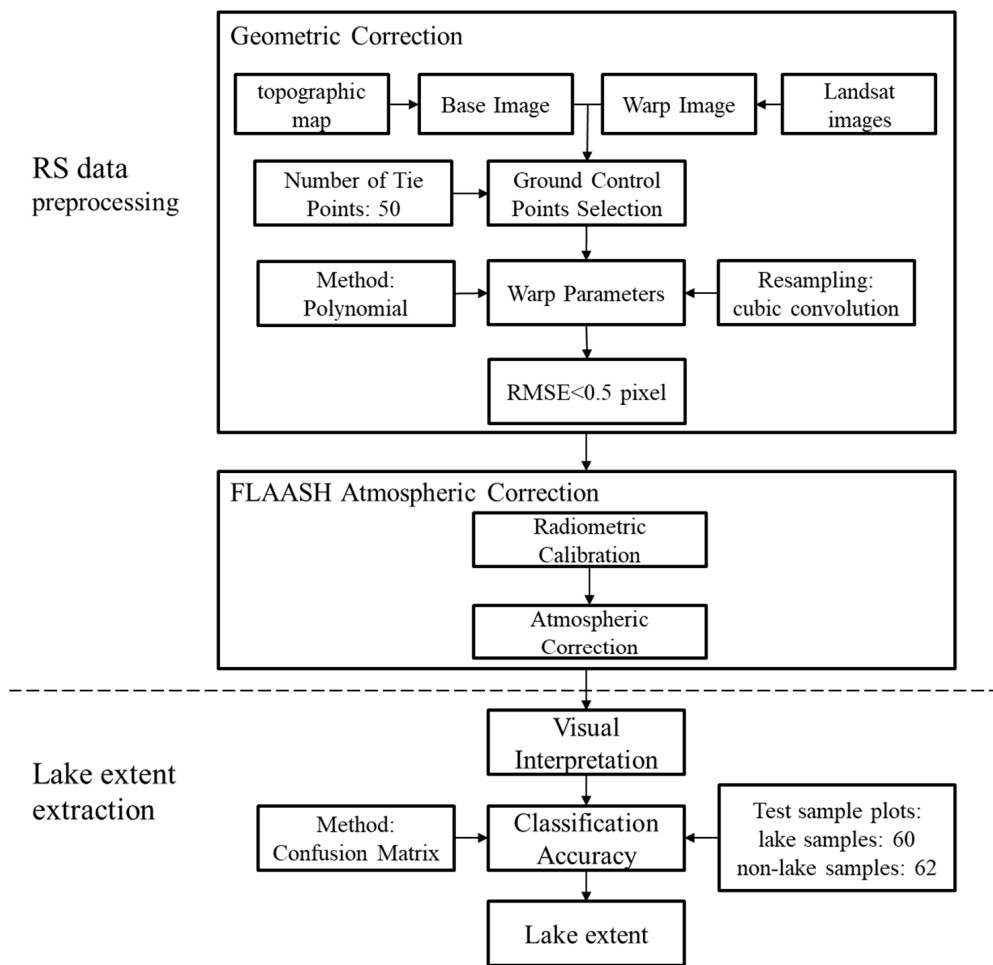

**Figure 2.** Workflow of lake area mapping.

After the steps mentioned above, 30 Landsat satellite images were used for the extraction of the extent of lakes. Then, we used the method of visual interpretation to delineate the selected lakes and non-lake land covers between 1972 and 2021 (Figure 2). Accuracy of lake extent was assessed by using the method of Confusion Matrix [55]. In this study, the overall accuracy of Lake LongmuCo and Lake Jiezechaqia extents in 2021 were 99.30% and 99.09%, k = 0.9826 and 0.9789; producer accuracies were 99.44% and 100.00%; and user accuracies were 98.01% and 97.21%, respectively.

### 2.2.6. Data Analysis

The equations used to calculate the changes in the lake area are as follows [56–58].

$$\alpha = \frac{S_{next} - S_{previous}}{S_{previous}} * 100\% \tag{1}$$

$$\beta = \frac{S_{next} - S_{previous}}{S_{previous}} * \frac{1}{T} * 100\% \tag{2}$$

In Equations (1) and (2), $\alpha$ (%) is the proportional change in the lake area and $\beta$ (%) is the annual change rate. $S_{previous}$ (km$^2$) is the lake area in the previous period, $S_{next}$ (km$^2$) is the lake area in the next time slot, and $T$ (a) is the period in years. In this study, the above equations were used to explore the area changes of Lake LongmuCo, Lake Jiezechaqia, Artificial Salt Lake 1, and Artificial Salt Lake 2, respectively.

When analyzing the impacts of climate factors on lake area changes, a large number of studies used Pearson's correlation coefficient [31,58]. We also used this coefficient for analyzing environmental impacts on lake changes and for comparison between our results and other datasets.

The least squares method was often used in numerous studies to detect the overall trends and to divide a time series into different subperiods by providing several trend change points [59,60]. In this study, we used the operation code outlined by Tomé and Miranda [59] in the ENVI 5.3/IDL 8.5 software to explore the stage characteristics of lakes and climate factors.

## 3. Results

### 3.1. Pattern of Lake Changes

From 1972 to 2021, the average area of Lake LongmuCo was 106.80 km$^2$. The area of Lake LongmuCo was 100.60 km$^2$ in 1972 and rose to 113.48 km$^2$ in 2021 (Table S1), and it increased by 12.88 km$^2$ from 1972 to 2021. Over the same period, the proportional change in the area of Lake LongmuCo, $\alpha$ (%), was 12.81% (r = 0.929, $p$ = 0.000 < 0.01) (Figure 3). This means that the annual rate of change in the lake area $\beta$ (%), was 0.26%/a. Figure 3 shows that changes in Lake LongmuCo could be divided into two periods: (1) a slight expansion from 1972 to 1999; (2) a period of rapid expansion from 2000 to 2021. During 1972–1999, the area of Lake LongmuCo increased with a $\beta$ (%) of 0.09%/a (r = 0.872, $p$ = 0.005 < 0.01), while it increased with a $\beta$ (%) of 0.43%/a (r = 0.998, $p$ = 0.000 < 0.01) during 2000–2021.

During 1972–2021, the average area of Lake Jiezechaqia was 114.98 km$^2$. The area of Lake Jiezechaqia was 108.35 km$^2$ in 1972 and rose to an expanded 122.31 km$^2$ in 2021 (Table S1), and it increased by 13.96 km$^2$ from 1972 to 2021. Over the same period, $\alpha$ (%) of Lake Jiezechaqia was 12.88% (r = 0.924, $p$ = 0.000 < 0.01) (Figure 3), and this means that $\beta$ (%) was 0.26%/a. Figure 3 shows that changes in Lake Jiezechaqia also could be divided into two periods. During 1972–1999, a slight expansion occurred in Lake Jiezechaqia with a $\beta$ (%) of 0.07%/a (r = 0.795, $p$ = 0.058 < 0.1). During 2000–2021, a rapid expansion occurred in Lake Jiezechaqia with a $\beta$ (%) of 0.47%/a (r = 0.992, $p$ = 0.000 < 0.01), and the change rate of lake area in this period is similar to that of Lake LongmuCo, which is obviously higher than the change rate of 1972–1999.

### 3.2. Environmental Impacts on Lake Changes

From 1972 to 2021, a significant increase in annual mean temperature at a rate of 0.05 °C/a (r = 0.799, $p$ = 0.000 < 0.01) demonstrated a progressively warmer climate throughout the basins of Lake LongmuCo and Lake Jiezechaqia and their surrounding areas (Figure 4). As shown in Figure 4, it can be seen that the annual precipitation of the study area fluctuated in the period of 1972–2021 (r = 0.031, $p$ = 0.830 > 0.1). Over the past 50 years, the annual mean temperature and annual precipitation varied from −0.16 °C and 70.0 mm to 2.33 °C and 84.7 mm, respectively. During 1972–2021, the annual mean temperature reached a maximum in 2016 (3.65 °C), while the highest annual precipitation occurred in 2015 (138.2 mm). Furthermore, in the 1993–2021 period, the increased rate of annual mean temperature (0.06 °C/a, r = 0.598, $p$ = 0.000 < 0.01) was twice as high as that (0.03 °C/a, r = 0.302, $p$ = 0.092 < 0.1) in the period of 1972–1992.

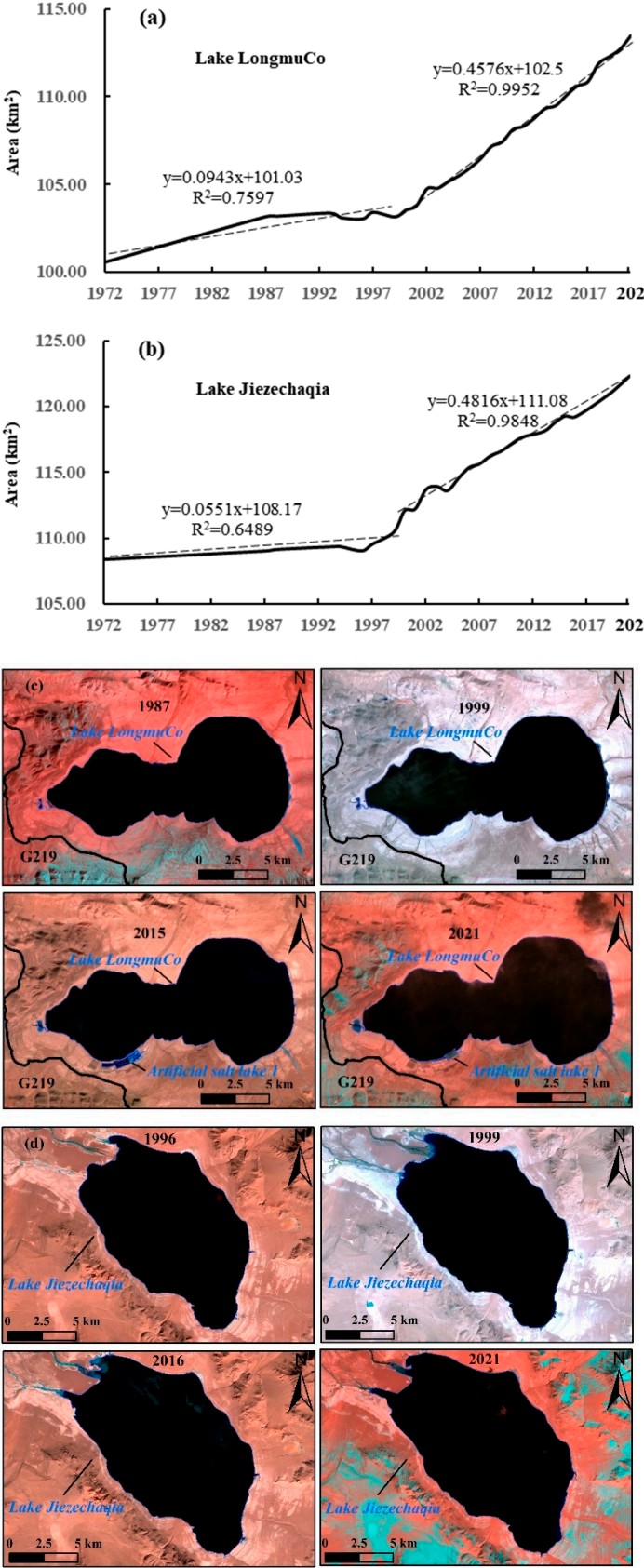

**Figure 3.** Area changes (**a**,**b**) and spatial distribution in different years (**c**,**d**) of Lake LongmuCo and Lake Jiezechaqia between 1972 and 2021.

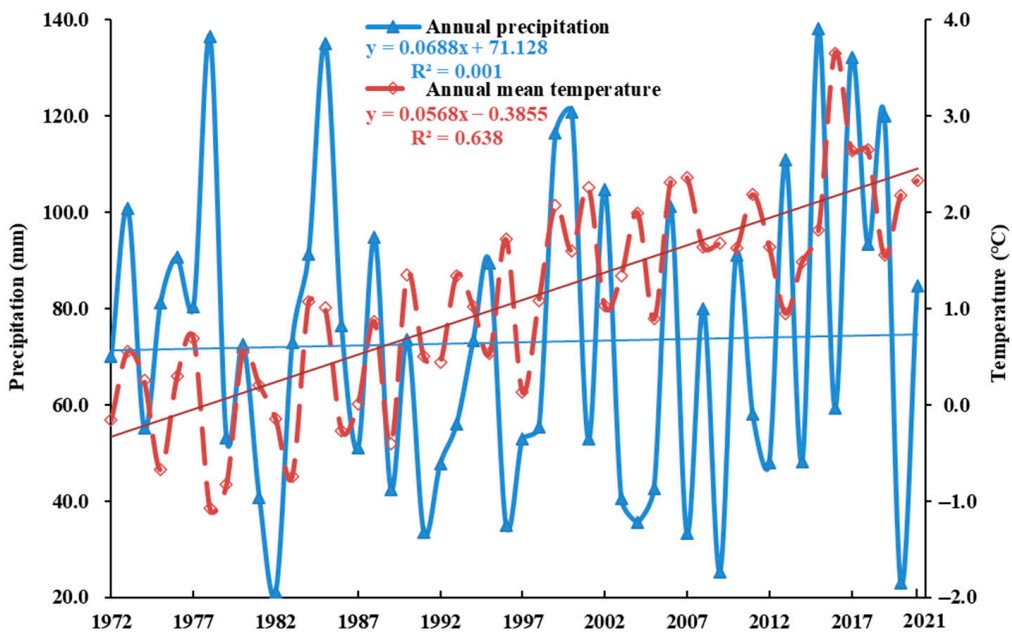

**Figure 4.** Changes in the annual mean temperature and annual precipitation around the study area during 1972–2021.

The results (Table 2) indicated positive correlations between temperature and areas of Lake LongmuCo and Lake Jiezechaqia, reaching the 99% confidence level. The good correlation implied that the annual mean temperature was likely a major contributor to the observed changes in the areas of Lake LongmuCo and Lake Jiezechaqia from 1972 to 2021. Table 2 also revealed that precipitation was not correlated with the areas of Lake LongmuCo and Lake Jiezechaqia. Furthermore, Lake LongmuCo and Lake Jiezechaqia are glacier-fed lakes [10,61] and the glaciers around these two lakes decreased by 21.81% in the first and second glacier inventories of China (Figure 5). The area of the largest glacier (Glacier Code: 5Z431I0075) around Lake LongmuCo and Lake Jiezechaqia decreased by 43.26%, while that of the smallest glacier (Glacier Code: 5Z431I0049) near these two lakes decreased by 95.53%. It seems that the increase in water supply from warming-triggered glacier melting was a reason for expansion of Lake LongmuCo and Lake Jiezechaqia [49].

**Table 2.** Pearson's correlation coefficients for lake area and climate factors.

| Period | Lake | Annual Mean Temperature | Annual Precipitation |
|---|---|---|---|
| 1972–2021 | LongmuCo | 0.588 ** | 0.180 |
| 1972–2021 | Jiezechaqia | 0.502 * | 0.124 |

Notes: ** $p < 0.01$, * $p < 0.05$.

### 3.3. Human Impacts on Lake Changes

The TP reserves comprise 75% of China's lithium resources, primarily distributed in salt lake brine. In recent years, increasing numbers of salt lakes have been mined to obtain lithium resources, although these alpine lakes are very sensitive to human activities [9]. The extensive exploitation has resulted in severe changes in the lake area, lake function, and landscape of the TP [9,62]. The Lake LongmuCo and Lake Jiezechaqia reserves comprise 1.89 and 2.01 million tons of lithium resources, respectively [49]. After 2013, salts from the Lake LongmuCo and Lake Jiezechaqia began to be mined and industrialized (Figure 6).

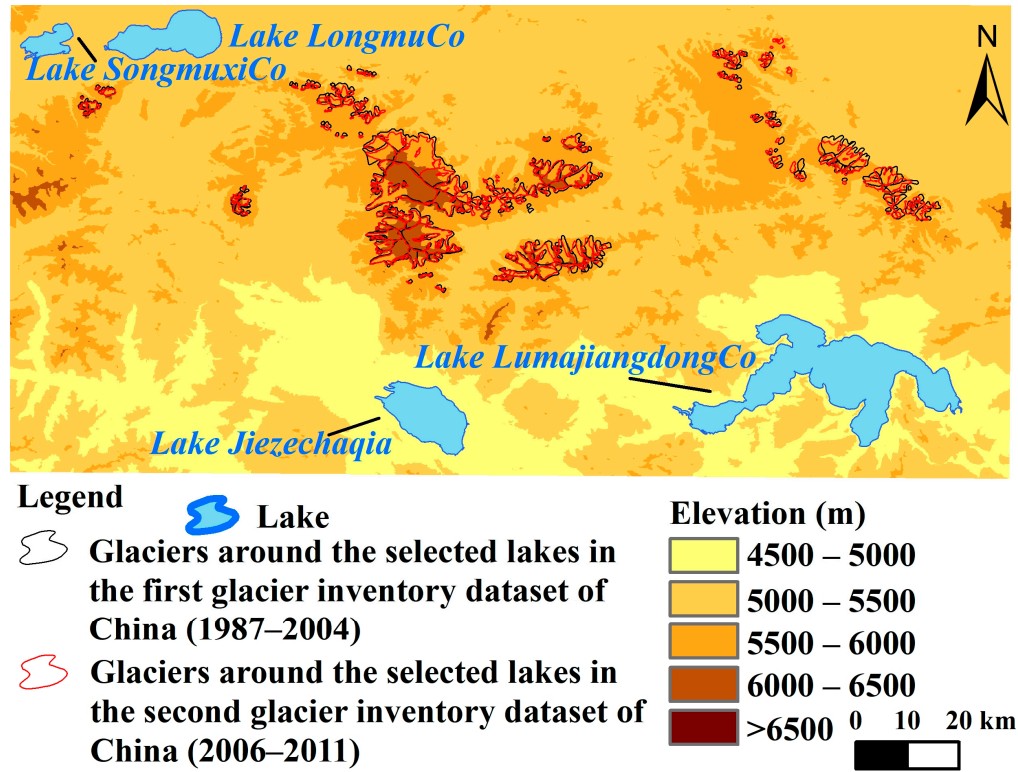

**Figure 5.** Area changes of glaciers around Lake LongmuCo and Lake Jiezechaqia.

More specifically, Figure 6 illustrated the changes in Lake LongmuCo and Lake Jiezechaqia exposed to the exploitation of lithium resources. Since 2013, one state-owned enterprise engaging in Lake LongmuCo exploitation has settled in and built dams to shift the water from Lake LongmuCo to an artificial salt lake (Figure 6). From 2013 to 2021, the average area of the Artificial Salt Lake 1 constructed by the state-owned enterprise was 0.88 km². The area of this artificial salt lake was 0.24 km² in 2013 and rose to 0.51 km² in 2021 (Table S1), and the maximum area of it occurred in 2015 (1.49 km²). Figure 6 shows that the changes in the Artificial Salt Lake 1 can be divided into two periods: (1) a rapid expansion from 2013 to 2015; (2) a decrease from 2016 to 2021. During 2013–2015, the area of Artificial Salt Lake 1 increased, with a $\beta$ (%) of 174.96%/a (r = 0.928, *p* = 0.011 < 0.05). During 2016–2021, the area of Artificial Salt Lake 1 decreased, with a $\beta$ (%) of −9.36%/a (r = −0.973, *p* = 0.003 < 0.01).

In 2013, the enterprise that mined Lake LongmuCo also industrialized Lake Jiezechaqia and shifted water from Lake Jiezechaqia to Artificial Salt Lake 2 through a 20 km long pipeline. Different from the exploitation of Lake LongmuCo, Artificial Salt Lake 2 was not adjacent to Lake Jiezechaqia, and lay to the northwest of Lake Jiezechaqia. During 2013–2021, the average area of the Artificial Salt Lake 2 was 8.79 km² (Figure 6). The area of this artificial salt lake was 2.67 km² in 2013 and rose to 9.80 km² in 2021 (Table S1), and the maximum area of it occurred in 2017 (12.02 km²). Figure 6 shows that changes in Artificial Salt Lake 2 also could be divided into two periods. During 2013–2017, a rapid expansion occurred in Artificial Salt Lake 2 with a $\beta$ (%) of 70.04%/a (r = 0.940, *p* = 0.017 < 0.05). During 2018–2021, a slight decrease occurred in Artificial Salt Lake 2 with a $\beta$ (%) of −3.69%/a (r = −0.702, *p* = 0.149 > 0.1).

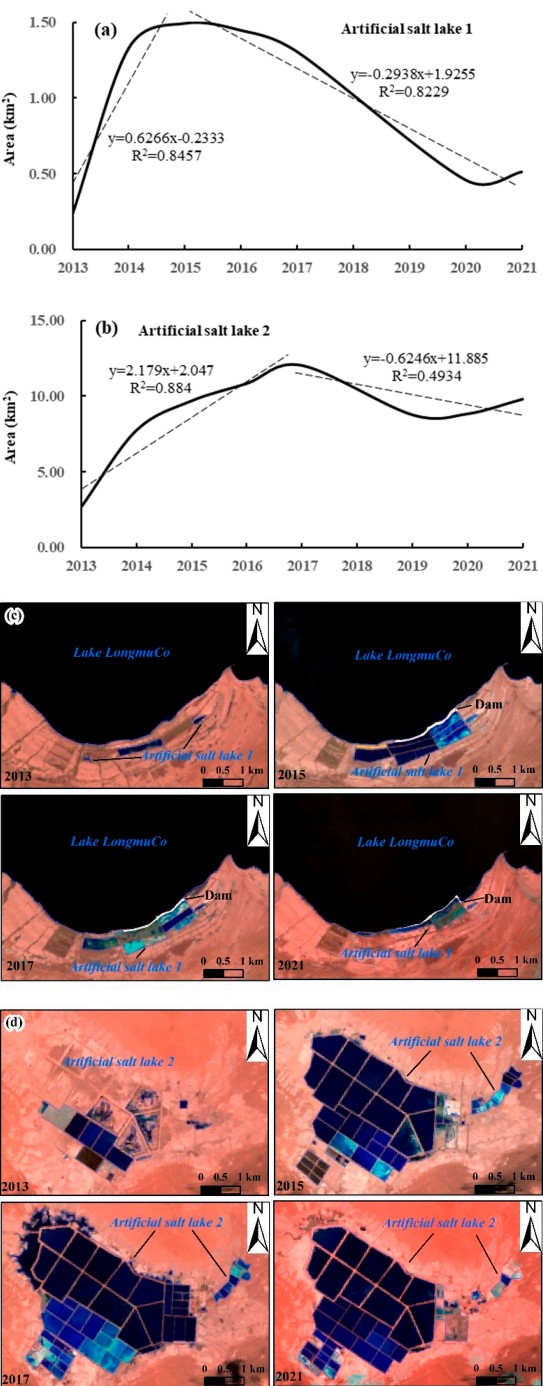

**Figure 6.** Area changes (**a**,**b**) and spatial distribution in different years (**c**,**d**) of the artificial salt lakes between 2013 and 2021.

Although the areas of the two artificial salt lakes have been shrinking in recent years, salt lake exploitations have spawned a host of complex environmental issues in the salt lake regions. Here, we selected two lakes (Lake SongmuxiCo and Lake LumajiangdongCo) in the CNNR to compare their area changes with that of Lake LongmuCo and Lake Jiezechaqia. These two lakes, like Lake LongmuCo and Lake Jiezechaqia, are also endorheic lakes and glacier-fed lakes [54]. The above four lakes could be observed on the same Landsat image. Among them, Lake SongmuxiCo is adjacent to Lake LongmuCo and Lake Lumajiang-dongCo is adjacent to Lake Jiezechaqia (Figure 5). Area changes of Lake SongmuxiCo and Lake LumajiangdongCo were mainly affected by climate change over the past several

decades [41–44,54], and according to Landsat image monitoring in this study, there was no exploitation of salt lakes in these two lakes in recent years. Figure 7 illustrated the area and water storage changes of Lake LongmuCo, Lake Jiezechaqia, Lake SongmuxiCo, and Lake LumajiangdongCo. From 1972 to 2021, the above four lakes all experienced lake expansion, Lake SongmuxiCo and Lake LumajiangdongCo expanded by more than 20%, while Lake LongmuCo and Lake Jiezechaqia only expanded by ~13% (Figure 7). After 2013, the $\beta$ (%) of Lake LongmuCo and Lake Jiezechaqia was less than 0.5%/a, while the $\beta$ (%) of Lake SongmuxiCo and Lake LumajiangdongCo was more than 1%/a (Figure 7). It seems that shifting the water from Lake LongmuCo and Lake Jiezechaqia to artificial salt lakes for extracting lithium has retarded the rate of expansion of these two lakes. During 2015–2019, when the salt lakes were developed, the change rates of water storage in Lake LongmuCo and Lake Jiezechaqia decreased by 7.68% and 40.05% compared with 2010–2015 (Figure 7). At the same time, the change rates of water storage in Lake SongmuxiCo and Lake LumajiangdongCo, which have no salt lake exploitations, increased by 48.44% and 6.95% compared with 2010–2015 (Figure 7). Then, in the period of 2017–2021, Lake LongmuCo inundated its south lakeshore outside the dams during its southward expansion (Figure 6). In other words, the dams that created Artificial Salt Lake 1 have blocked the expansion of Lake LongmuCo to the south (Figure 6). Next, the artificial salt lake exploitations have altered the landscape surface from grassland and saline alkali land to a water body [63]. Due to the vulnerability of the environment within Lake LongmuCo and Lake Jiezechaqia, as well as human intervention, the environmental processes of these lakes have been strongly influenced.

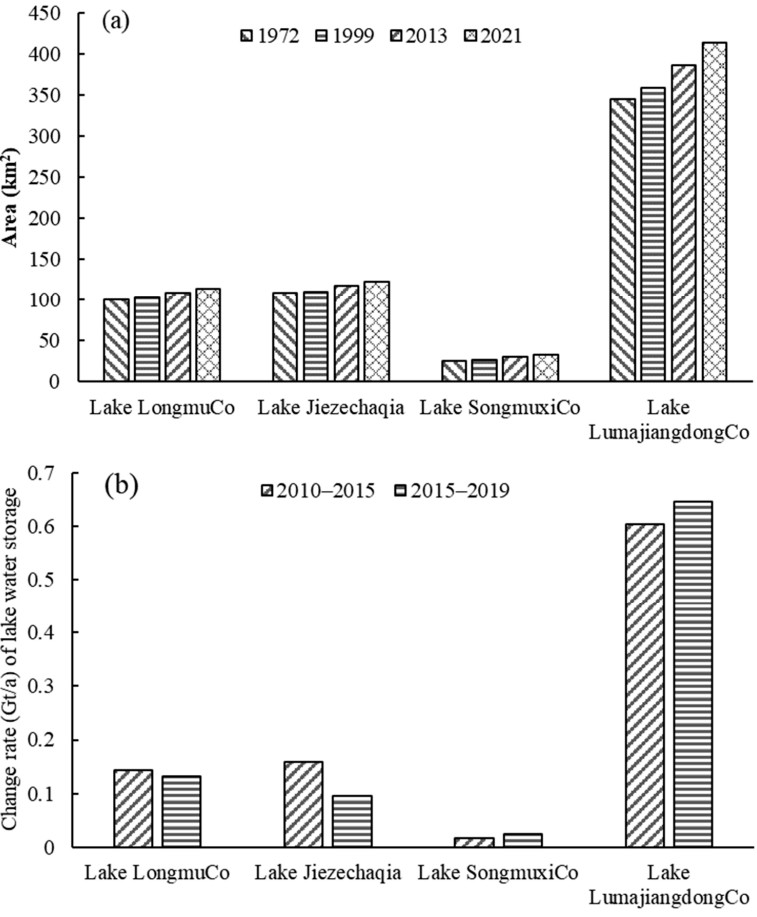

**Figure 7.** Comparison of area (**a**) and water storage (**b**) changes of Lake LongmuCo, Lake Jiezechaqia, Lake SongmuxiCo, and Lake LumajiangdongCo.

## 4. Discussion

Lakes are important indicators of the Earth's hydrological cycle [64]. A large number of studies have revealed that lake changes over the TP were characterized by strong spatio-temporal heterogeneity during the 1960s–2021 [10,35,65–69]. A long-term dataset for lakes on the TP was created by Zhang et al. [10] using Landsat imagery, and we compared our results with this dataset. Note that for the areas of Lake LongmuCo and Lake Jiezechaqia, the data obtained by Zhang et al. [10] and the data in this study were consistent (r = 0.996 and 0.999, *p* = 0.000 < 0.01). This means that the results of this study are credible. The area of Lake LongmuCo increased at a rate of 0.22%/a during 1970–2021 [10], and we found that the area of Lake LongmuCo increased at a rate of 0.26%/a during the same period. Our study confirmed the findings of Zhang et al. [10].

Over the past decades, lake expansion, especially in the Inner TP, has become one of the most prominent environmental change events on the TP [26,35]. Approximately three-quarters of the lakes on the TP showed noticeable expansion since the late 1990s and accelerated their expansion in the 2000s [38,68]. The year 2000 appeared to be a trend changing point for lake area changes on the TP [57]. We also found that Lake LongmuCo and Lake Jiezechaqia exhibited remarkable growth after 1999. Our study corroborated the findings of previous studies [35–38,66].

Over the TP, the annual mean temperature noticeably increased at a rate of 0.32 °C/decade since 1961 [67]. We found that the annual mean temperature significantly increased at a rate of 0.50 °C/decade since 1972 in the study area, and it was higher than that of the TP.

Numerous previous studies have shown that climate factors such as precipitation, warming-triggered glacial meltwater, air temperature, evapotranspiration, and warming-triggered permafrost melting were important contributors to lake changes over the TP in recent decades [10,24–26,34–39,63–66]. Moreover, the temperature was likely responsible for the observed lake changes on the Changtang Plateau [29,38,57,65]. We also found that Lake LongmuCo and Lake Jiezechaqia within the Changtang Plateau were sensitive to air temperature. Some researchers have noticed that there appears to be a positive spatial correlation between glacier melting and lake expansion [49,57,68,69]. We also found that Lake LongmuCo and Lake Jiezechaqia may receive an increased meltwater supply from glacier melting under the influence of warmer climates.

The natural ecosystems of the TP are very sensitive to human intervention [8,26,29,46,70], and the establishment of nature reserves has acted as an effective strategy for conserving ecosystems and restricting and prohibiting human activities [25–30]. The protected areas include various types of ecosystems, such as lakes, grasslands, and forests. Although the vast majority of lakes on the TP have been protected, there are still some lakes that have been developed by humans for economic profit [9,25]. A recent study indicated that the exploitation of salt lakes created a threat to the environment of the TP [9]. The dams constructed by industrial enterprises have altered water flow, affected the direction of lake changes, and disturbed local ecosystems [9,71]. In this study, we found that the artificial salt lake exploitations in the regions of Lake LongmuCo and Lake Jiezechaqia have retarded the rate of expansion of these two lakes during 2013–2021, and found that the dams of the artificial salt lake have blocked the expansion of Lake LongmuCo to the south.

## 5. Conclusions

This study used long-term Landsat imagery and meteorological records to monitor the temporal evolution patterns of two major lakes in the Changtang National Nature Reserve during 1972–2021 and examined climatic and anthropogenic impacts on lake area changes. Over nearly 50 years, the area of Lake LongmuCo and Lake Jiezechaqia significantly increased by 12.88 km$^2$ and 13.96 km$^2$, respectively. After 1999, Lake LongmuCo and Lake Jiezechaqia all entered into a period of rapid expansion. The annual mean temperature has been increasing since 1972 at a rate of 0.05 °C/a, while the change in annual precipitation was not significant. Furthermore, the annual mean temperature noticeably increased

at a rate of 0.06 °C/a during 1993–2021. Temperature was a major contributor to the observed lake changes over the basins of Lake LongmuCo and Lake Jiezechaqia between 1972 and 2021. And the glaciers around these two lakes decreased by 21.81%, the increase in water supply from warming-triggered glacier melting was a reason of expansion of Lake LongmuCo and Lake Jiezechaqia. With the mining exploitations of salt lakes since 2013, human activities directly contributed to the changes of Lake LongmuCo and Lake Jiezechaqia during 2013–2021. The areas of the two artificial salt lakes affiliated to Lake LongmuCo and Lake Jiezechaqia were 0.24 km$^2$ and 2.67 km$^2$ in 2013 and rose to 0.51 km$^2$ and 9.80 km$^2$ in 2021, respectively. Additionally, the maximum area of the two artificial salt lakes occurred in 2015 (1.49 km$^2$) and in 2017 (12.02 km$^2$), respectively. It seems that shifting the water from Lake LongmuCo and Lake Jiezechaqia to artificial salt lakes for extracting lithium has retarded the rate of expansion of these two lakes. The dams constructed by industrial enterprises have blocked the expansion of Lake LongmuCo to the south. These findings provide vital new light on the responses of lakes to climate change and anthropogenic activities. Now we need more comprehensive and in-depth efforts to protect lake ecosystems within the national nature reserves.

**Supplementary Materials:** The following supporting information can be downloaded at: https://www.mdpi.com/article/10.3390/land12020267/s1, Table S1: The measured lake areas in this study.

**Author Contributions:** Conceptualization, Z.Z. and Z.H.; methodology, Z.Z. and Z.H.; software, Z.Z. and J.Z.; validation, Z.H. and R.K.; formal analysis, Z.Z.; investigation, Z.Z. and Z.H.; resources, Z.Z.; data curation, Z.H.; writing—original draft preparation, Z.Z., Z.H., W.L. and J.Z.; writing—review and editing, Z.Z., Z.H., W.L. and R.K.; visualization, Z.Z. and Z.H.; supervision, Z.Z.; project administration, Z.Z.; funding acquisition, Z.Z. and Z.H. All authors have read and agreed to the published version of the manuscript.

**Funding:** This research was funded by the National Natural Science Foundation of China, grant number 42101293, and the Major Program of National Social Science Foundation of China, grant number 17ZDA158.

**Data Availability Statement:** The data presented in this study are available on request from the corresponding author.

**Conflicts of Interest:** The authors declare no conflict of interest.

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
