# Peer review of "Response of Two Major Lakes in the Changtang National Nature Reserve, Tibetan Plateau to Climate and Anthropogenic Changes over the Past 50 Years"

_land, doi:10.3390/land12020267_

Round 1

Reviewer 1 Report (Previous Reviewer 1)

I have previously reviewed this paper and I find authors tried to improve the manuscript. However, reading the revised version also it is regretful to say that the paper is too general and does not advance our knowledge. 

Specifically, the linkage between anthropogenic and climatic driver of lake changes are not satisfactory. Paper needs more data, deeper analysis and should somewhat display new knowledge.

Author Response

Reviewer 2 Report (Previous Reviewer 2)

I appreciate the efforts of the authors to revise the paper. Now, this could be published as a paper.

Author Response

Reviewer 3 Report (Previous Reviewer 3)

I think the previous comments and suggestions have been answered and I recommend to accept the manuscript in present form.

Round 2

Reviewer 1 Report (Previous Reviewer 1)

Paper is improved than previous version.

This manuscript is a resubmission of an earlier submission. The following is a list of the peer review reports and author responses from that submission.

Round 1

Reviewer 1 Report

Refer to attached pdf for peer-review report.

Reviewer 2 Report

General comments

This paper tried to reveal the response of two lakes and nearby salt lakes to climate change and anthropogenic forcing using RS data. This might be interesting if adequately analyzed. The current manuscript combines two reports of areal changes for the natural lakes and for the salt lakes. The area change should be analyzed and discussed regarding the water budget. The reviewer suggests that authors should resubmit with sound analysis. Please see the specific comments below.

Specific comments

1.   L. 27: sheds -> shows

2. Introduction: Most of the references are irrelevant. The research focus of this study is quite unclear.

3. L. 66-76: Why two lakes are targeted? The reason of the limited studies is not a good explanation. 

4. Result: 

  For both Lake LongmuCo and Lake Jiezechaquia, different trends of areas are seen in 50 years. Why this occurred? Need some analysis. How the impact of temperature rise differs between a glacier-fed lake and non-glacier-fed lake? L.208-212 needs some data.

  For two salt lakes, there is no analysis for the different trend of areas.

5. L.226: Fig 5 also shows the exploitation area of lithium?

Reviewer 3 Report

Your paper is well written but has minor editing problems. Please revise lines 153-155